# Changes in Nitrogen Pools in the Maize—Soil System after Urea or Straw Application to a Typical Intensive Agricultural Soil: A ¹⁵N Tracer Study

**Jie Zhang [1,2], Ping He [2], Dan Wei [3], Liang Jin [3,4], Lijuan Zhang [1], Ling Li [5], Shicheng Zhao [2], Xinpeng Xu [2], Wei Zhou [2] and Shaojun Qiu [2,\*]**

[1] College of Resources and Environmental Sciences, Hebei Agricultural University, Baoding 071000, China; jiezhang0320@126.com (J.Z.); lj_zh2001@163.com (L.Z.)
[2] Key Laboratory of Plant Nutrition and Fertilizers, Ministry of Agriculture and Rural Affairs, Institute of Agricultural Resources and Regional Planning, Chinese Academy of Agricultural Sciences, Beijing 100081, China; phe@ipni.net (P.H.); zhaoshicheng@caas.cn (S.Z.); xuxinpeng@caas.cn (X.X.); zhouwei02@caas.cn (W.Z.)
[3] Institute of Plant Nutrient and Resources, Beijing Academy of Agriculture and Forestry Sciences, Beijing 100097, China; wd2087@163.com (D.W.); jinliang19762003@yeah.net (L.J.)
[4] Institute of Soils, Fertilizers and Environmental Resources, Heilongjiang Academy of Agricultural Sciences, Harbin 150086, China
[5] China Agricultural Science and Technology Press, Chinese Academy of Agricultural Sciences, Beijing 100081, China; lling7856@163.com
\* Correspondence: qiushaojun@caas.cn; Tel.: +86-10-82105029; Fax: +86-10-82106225

**Abstract:** A maize pot experiment was conducted to compare the difference of N distribution between bulk and rhizospheric soil after chemical fertilizer with or without soil straw amendment at an equivalent N rate using a ¹⁵N cross-labeling technique. Soil N pools, maize N and their ¹⁵N abundances were determined during maize growth. The urea plus straw treatment significantly ($p < 0.05$) increased the recovery of urea N in soil and 26.0% of straw N was assimilated by maize. Compared with urea treatment in bulk soil, urea plus straw treatment significantly ($p < 0.05$) increased the concentration and percentage of applied N as dissolved organic N (DON) and microbial biomass N (MBN) from milk stage to maturity, increased those as particulate organic N (PON) and mineral associated total N (MTN) throughout maize growth and decreased those as inorganic N (Inorg-N) from the eighth leaf to the silking stage. Compared with bulk soil, rhizospheric soil significantly ($p < 0.05$) decreased the concentration and percentage of applied N as PON and increased those as Inorg-N and MTN in both applied N treatments from the silking stage, and significantly ($p < 0.05$) decreased the concentration and percentage of applied N as microbial biomass N (MBN) in the urea plus straw treatment. Overall, straw N was an important N source and combined application of chemical fertilizer with straw increased soil fertility, with the rhizosphere regulating the transformation and supply of different N sources in the soil–crop system.

**Keywords:** bulk soil; rhizosphere soil; soil N pools; maize N uptake; pot experiment

## 1. Introduction

Nitrogen (N) has made a large contribution to crop production to meet the food demand of an increasing world population since World War II [1]. However, long-term uncontrolled chemical fertilizer N application has aggravated soil acidification, groundwater eutrophication, and greenhouse gas emissions [2–4]. Moreover, large amounts of straw are generated as a result of the high grain yields produced. Straw burning leads to atmospheric particulate and oxynitride pollution. Instead, the combined application of chemical fertilizer and straw at controlled rates may be an efficient method

of alleviating these detrimental effects and increasing soil nutrient status [5,6], although small amounts of greenhouse gas emissions might be stimulated [7–9]. In addition, straw incorporation provides a useful source of N and also provides carbon (C) sources for soil microbes, affecting the distribution of N in the crop–soil system [10,11]. Thus, understanding the distribution of chemical fertilizer- and straw-N in crop–soil systems is necessary to evaluate the application of chemical fertilizer combined with straw.

The rhizosphere is a zone of high nutrient content for soil microorganisms and roots [12]. Relative to the bulk soil the higher microbial biomass in the rhizosphere results in higher competition for N [13]. When large amounts of easily available C derived from root exudates in the rhizosphere, this situation promotes microbial N immobilization which contributes to the increasing competition for N [14,15]. On the other hand, substantial crop N assimilation generates relative N deficiency in the rhizosphere during crop growth [16]. This strong depletion, caused by competition between roots and microbes, promotes the transport of N into the rhizosphere from the bulk soil [17]. However, this transportation is closely correlated with the formations of N, e.g., $NO_3^-$ diffuses rapidly, while $NH_4^+$ diffuses slowly [18,19]. All of the above factors make the transformation of N more complicated in the rhizosphere than in the bulk soil. Therefore, a comparison of N behavior in bulk and rhizospheric soils is necessary to understand how the crop rhizosphere affects changes in soil N during crop growth.

Chemical fertilizer N application can stimulate soil organic matter mineralization and promote soil N transformation among the different soil N pools in agricultural systems [20]. Correspondingly, soil N can be divided into labile and stable pools. The labile N pools, easily decomposed by microbes, are highly sensitive to fertilizer management and the stable pools promote soil N retention [21]. The labile soil N pools are separated from soil inorganic N (Inorg-N), microbial biomass N (MBN), dissolved organic N (DON), and particulate organic N (PON) [20]. Inorg-N, the main source of N to crops, is derived mainly from chemical fertilizers and can be immobilized into or released from organic N pools in agricultural soils [20], and MBN regulates the immobilization of Inorg-N and the mineralization of soil organic N. DON can be directly or indirectly assimilated by crops and microorganisms, and DON is derived from microbial metabolism and root exudation and can be further decomposed to Inorg-N via microbial activity [22,23]. PON represents partly decomposed plant residues and is an important N source for soil microorganisms [24]. When particulate organic matter is removed from the whole soil through wet sieve, the residual N is regarded as mineral-associated total N (MTN). MTN holds the largest proportion of the soil total N [25] and retains N substrates by sorption on mineral surfaces or the formation of organic–mineral complexes [26,27]. These N pools can therefore reflect soil biological, chemical, and physical properties. Thus, the investigation of these soil N pools may provide important information on soil N transformations.

Straw returned to the soil provides C sources for microorganisms and increases soil Inorg-N immobilization [28,29]. Previous studies show that the combined application of chemical fertilizer with straw increased the size of the MBN pool [30,31]. Bai et al. [32] reported that chemical fertilizer combined with straw return increased the contribution of applied N to MBN, PON, and MTN compared with chemical fertilizer alone. Wang et al. [33] found that application of organic materials greatly increased the concentration of DON in comparison with chemical fertilizer applications. Overall, these studies have focused mainly on changes in the distribution of applied N in soils without plants with equivalent N rates. However, competition for N occurs between microorganisms and crops to greatly alter the retention of N in different soil N pools.

Here, we quantified the effect of applied N in plant (shoots, grains, roots) and soil N (Inorg-N, MBN, DON, PON, MTN) pools with chemical fertilizer application with or without straw amendment, and compared the difference of urea N and straw N in different N pools in bulk and rhizospheric soils under chemical fertilizer application with or without straw amendment during the growth of maize. We hypothesized that a portion

of straw N can be assimilated by maize, and that the transformation of organic N would be accelerated in the rhizosphere. A pot experiment was conducted with $^{15}$N-labeled urea and $^{15}$N-labeled maize straw in order to address these issues.

## 2. Materials and Methods

### 2.1. Soil

Soil samples were collected from an arable field (0–20 cm depth) at Heilongjiang Academy of Agricultural Sciences near Harbin city, Heilongjiang province, northeast China (45°50′ N, 126°51′ E). The soil is classified as a Mollisol according to the USDA classification system [34]. This region is characterized by low temperatures and a continental monsoon climate with an average annual precipitation of 486 mm, mean annual temperature of 3.6 °C, and mean monthly temperature of 19.3 °C from May to September during crop growing season. Selected soil physicochemical properties are shown in Table 1.

**Table 1.** Selected physicochemical properties of the chernozem soil studied.

| Organic C (g kg$^{-1}$) | Total N (g kg$^{-1}$) | Olsen P (mg kg$^{-1}$) | NH$_4$OAc-Extractable K (mg kg$^{-1}$) | pH (units) | Soil Texture (%) | | |
|---|---|---|---|---|---|---|---|
| | | | | | Sand | Silt | Clay |
| 19.4 | 1.8 | 9.6 | 155.9 | 6.9 | 27.4 | 48 | 24.6 |

### 2.2. Preparation of Labeled and Unlabeled Straw

$^{15}$N-labeled straw was obtained by conducting a greenhouse pot experiment. As previously reported by Qiu et al. (2012) [5], labeled straw with a relatively high $^{15}$N abundance was produced by growing maize (cv. Zhengdan 958) in soil with 30.16 % $^{15}$N-labeled $(NH_4)_2SO_4$, $NaH_2PO_4$ and KCl at rates of 150 mg N, 65.5 mg P and 124.5 mg K per kg soil. To ensure sufficient soil N supplied, topdressing fertilizer was applied at the 8th leaf and tasseling growth stages of the labeled maize at rates of 1 and 2 g plant$^{-1}$ ($^{15}NH_4)_2SO_4$ at 30.16% $^{15}$N abundance, respectively. Maize grain formation was prevented by covering the corn cobs with paper bags before the silking stage. The unlabeled maize was grown in a field near the greenhouse with the same soil type and maize cultivar. The sowing and harvest dates of the unlabeled maize were the same as those of the labeled maize. After harvest the experimental straw materials were analyzed to determine their total C, N, P, and K concentrations and $^{15}$N abundance according to Qiu et al. [9]. In the labeled straw, total N concentration and $^{15}$N abundance were 9.6 ± 0.2 g kg$^{-1}$ and 15.24 ± 0.06%, total C, N, P, and K concentrations were 435.5 ± 1.2, 1.4 ± 0.02, and 11.2 ± 0.03 g kg$^{-1}$; in unlabeled straw, total N concentration and $^{15}$N abundance were 11.3 ± 0.2 g kg$^{-1}$ and 0.37 ± 0.0001%, total C, N, P, and K concentrations were 449.7 ± 0.17, 11.3 ± 0.2, 1.1 ± 0.01, and 14.2 ± 1.2 g kg$^{-1}$.

### 2.3. Experimental Design

An equivalent N amendment pot experiment with a completely randomized design was carried out in a greenhouse with a glass roof. The treatments were: (I) control (CK); (II) $^{15}$N-labeled urea (U); (III) urea plus straw (US). In order to distinguish the contribution of urea-N and straw-N to plant and soil N pools, two subtreatments were conducted in the US treatment: (1) $^{15}$N-labeled urea plus straw ($^{15}$U + S) and (2) $^{15}$N-labeled straw plus urea (U + $^{15}$S). In the US treatment, both of the subtreatments received the same management except for the difference of labeled N source, and the ratio of urea N to straw N was 6:4 according to Zhu and Wen [35]. Each 25-cm-diameter pot contained 10.0 kg air-dried soil uniformly treated with 1.5 g N, 0.655 g P$_2$O$_5$, and 1.245 g K$_2$O in all fertilizer treatments. In the controls the soil was treated with the same amounts of P and K fertilizers. The synthetic fertilizers used were urea, superphosphate, and potassium chloride. The $^{15}$N abundance of the urea was 15.10% which was produced by Shanghai Research

Institute of Chemical Industry in China. The ground straw was passed through a 0.25-mm sieve. In order to ensure that the N, P, and K application rates were equivalent in each pot, the contents of N, P, and K in the straw were included in the total fertilizer allocation. In order to explore N uptake and soil N transformation during whole maize growth, three replicate pots of each treatment were prepared to allow destructive sampling at the 8th leaf (V8), silking (R1), milk (R3), and physiological maturity (R6) growth stages, i.e., 35, 70, 90, and 111 days after the maize was sown, giving a total of 12 pots for each treatment.

Two maize seeds (cv. Zhengdan 958) were sown in the center of each pot and thinned to one plant after emergence. In order to prevent root growth along the inner wall of each pot, three PVC tubes (diameter 13.8 cm, height 35 cm) were inserted vertically into the soil. The base of each PVC tube was about 5 cm above the base of the pot and the PVC tubes were about 5 cm from the pot inner wall. Each PVC tube had three holes 8 cm from the base and the holes and the PVC tubes were enclosed with 0.5-mm mesh to prevent soil entering the PVC tube so that water in the tube flowed rapidly into the soil. Distilled water was added daily to each pot from the PVC tubes to constant weight, and then the top of each tube was sealed with a rubber stopper to prevent water loss. During maize growth the soil water content was adjusted daily to 60 % using the weight balance method.

At each stage the shoots (aboveground) and roots (belowground) were separated from the maize upper node brace root. During maize growth the "less green" bottom leaves were cut, oven-dried, stored, and then mixed into the same plant sample when the maize plant was taken, Then the sampled shoots, roots, and grains were oven-dried at 60 °C and weighed with a balance.

Rhizosphere and bulk soils were sampled as described by Peng et al. [36]. Briefly, the roots were removed from the pot and shaken to remove the loosely attached soil with roots, then the soil adhering to the root system was placed in a paper bag, vigorously shaken, and brushed to collect the closely adhering soil with roots. The soil adhering to the roots was regarded as rhizosphere soil and the remaining soil in each pot was mixed thoroughly and regarded as bulk soil. Any visible roots in either soil fraction were removed.

*2.4. Sample Analysis*

Soil organic C was analyzed with a CN analyzer (Macro cube, Elementar, Hanau, Germany). Soil Olsen-P was extracted with 0.5 M $NaHCO_3$ and determined at 880 nm , soil $NH_4OAc$-extractable K was extracted with 1 M $NH_4OAc$ and determined using atomic absorption spectrophotometry [37], and pH was measured in a 1:2.5 soil/water suspension (Thomas, 1996). Soil texture was determined with a laser particle size analyzer (LS13320, Beckman Coulter, Brea, CA, USA).

Total N in shoots, roots, grain, and soil was determined with an elemental analyzer (Macro cube, Elementar, Hanau, Germany). Plant samples were passed through a 0.25-mm sieve and soil samples through a 0.15-mm sieve. The $^{15}N$ abundance of shoot N, root N, grain N, and soil TN was determined using an isotope ratio mass spectrometer (Finnigan MAT251, Thermo Fisher, Waltham, MA, USA).

Soil Inorg-N was extracted with 1M KCl solution (1:5, *w/v*) on a reciprocal shaker for 1 h and then determined with a continuous flow analyzer (FIAstar 5000, FOSS, Hillerød, Denmark). For inorganic $^{15}N$ abundance the 10 mL KCl-extracted solution was reduced using Devarda's alloy and distilled. The distillates from the KCl extracts were quantified by titration and then acidified and oven-dried at 60 °C for N isotope analysis by mass spectrometry (Finnigan MAT251, Thermo Fisher, Waltham, MA, USA) according to Hauck et al. [38].

Soil MBN was determined using the $CHCl_3$ fumigation-$K_2SO_4$ method as described by Brookes et al. [39]. Briefly, fresh soil was fumigated with $CHCl_3$ for 24 h at 25 °C and then the N in fumigated and unfumigated samples was extracted with 0.5 M $K_2SO_4$ (1:4

*w/v*, 0.5 h). Fumigated and unfumigated solutions (20 mL) were analyzed using the Kjeldahl method. Soil microbial biomass N was calculated as the difference in N concentration between fumigated and unfumigated samples divided by a conversion coefficient of 0.45 [40]. Soil DON was calculated by subtracting Inorg-N from the N concentration in unfumigated solution extracted with 0.5 M $K_2SO_4$ [20,41]. The $^{15}$N abundance of fumigated and unfumigated solutions was determined as the $^{15}$N abundance of Inorg-N as described above.

Soil PON and MTN particles were fractionated as described by Bronson et al. [42]. Briefly, fresh soil at each growth stage, equivalent to 25 g oven-dried soil (<2 mm), was dispersed in 5% sodium hexametaphosphate solution (soil:solution ratio 1:4, *w/v*) on a reciprocal shaker for 60 min. The slurry was sieved using a 53-μm mesh until the deionized water became clear. The <53 and ≥53 μm soil fractions were transferred to beakers separately and oven-dried at 60 °C. Nitrogen in both isolated soil fractions is defined as PON (>53 μm) and MTN (<53 μm) and their N concentrations were determined using an elemental analyzer (Macrocube, Elementar, Hanau, Germany) after particulate organic matter and mineral associated matter were passed through a 0.15-mm sieve. The $^{15}$N abundance of PON and MTN was determined as total $^{15}$N as described above.

### 2.5. Calculations

The concentration of applied N present as MBN was calculated by the difference between the fumigated N concentration and the unfumigated N concentration derived from urea or straw and then divided by a conversion coefficient of 0.45. Correspondingly, DON derived from applied N was calculated by the difference between the unfumigated N content derived from urea or straw and the Inorg-N derived from urea or straw.

The concentrations and percentages of applied N in different soil N pools and plant parts were calculated using the following formula [32].

$$Ndfx_P = (CON_P \times APE_P)/APE_a$$

$$\text{Percentage of } Ndfx_P\ (\%) = Ndfx_P/CON_P \times 100$$

where Ndfx is N derived from urea or straw, CON is the concentration of N (mg N kg$^{-1}$), the subscript P is the soil or plant N pool, APE is $^{15}$N atom percent excess calculated by subtracting the $^{15}$N abundance of the control treatment from the applied N (straw N or urea N) treatments. The percentage of applied N in soil N pools and plant parts is the total N of each N pool present as urea N and straw N. Thus, in the US treatment, the percentage of applied N in each N pool is the sum of labeled straw N percentage in $^{15}$NU + S and labeled urea N percentage in $^{15}$NU + S in the target pool.

Here, the recovery rate is the percentage of applied N present in the target fraction at the harvest period [32].

### 2.6. Statistical Analysis

Data are expressed on oven-dried basis and presented as the mean of three replicates. One-way ANOVA statistical analysis of variance was conducted using the SPSS 16.0 software package (SPSS Inc., Chicago, IL, USA). Student's *t*-test was used at each growth stage to compare the differences in all variables of specific N pools between urea-only and urea plus straw treatments as well as between bulk and rhizospheric soils, and least significant difference (LSD) at the 0.05 protection level was used to assess the differences in mean values of the recovery rate among $^{15}$N-labeled urea, $^{15}$N-labeled urea plus straw and $^{15}$N-labeled straw plus urea treatments at maturity.

## 3. Results

### 3.1. Crop Biomass

From stages R1 to R6 (Figure 1), the significance ($p < 0.05$) of the shoot and grain values among the treatments was U > U + S > CK, and that of roots was U + S > U > CK except for R6 stage. The grain yields among CK, U, and U + S treatments were 33.6, 121.8 and 85.0 g plant$^{-1}$, respectively.

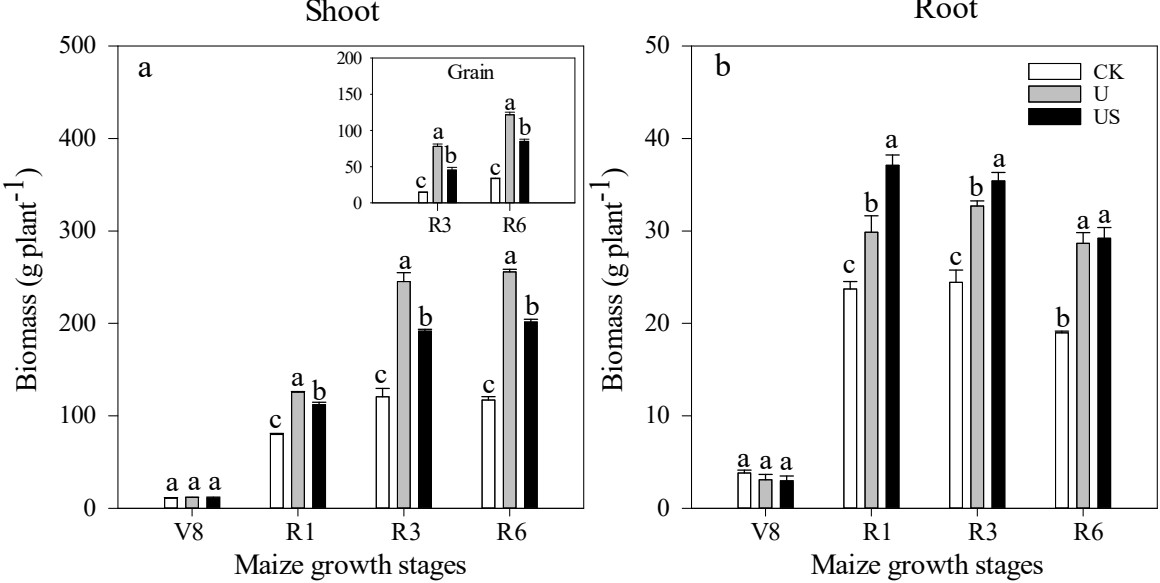

**Figure 1.** Biomass of maize shoots (**a**), grains, and roots (**b**) among control (CK), urea (U), and urea plus straw (US) treatments during maize growth stages of 8th leaf (V8), silking (R1), milk (R3), and physiological maturity (R6). Data shown are mean ± standard deviation of three replicates. Each replicate in US treatment was the mean of $^{15}$N labeled urea plus straw and $^{15}$N labeled straw plus urea treatments. Different letters indicate significant differences among treatments at $p < 0.05$. Shoot biomass includes leaf, stem, cob, husk, and grain fractions (all whole-plant fractions except roots).

### 3.1. Plant N Uptake

The accumulation of applied N in plant shoots, grains, and roots in the US treatment significantly ($p < 0.05$, Figure 2a,b) increased by 46.9–48.4%, 48.4–55.2%, and 24.6–27.7% relative to the U treatment except for shoots at stage V8. The percentages of applied N in shoots and roots in the US treatment were significantly ($p < 0.05$, Figure 2c,d) 12.3% and 12.0% higher than the U treatment at stage V8, and the opposite trend occurred in shoots, roots, and grains from stages R1 to R6 (Figure 2c,d).

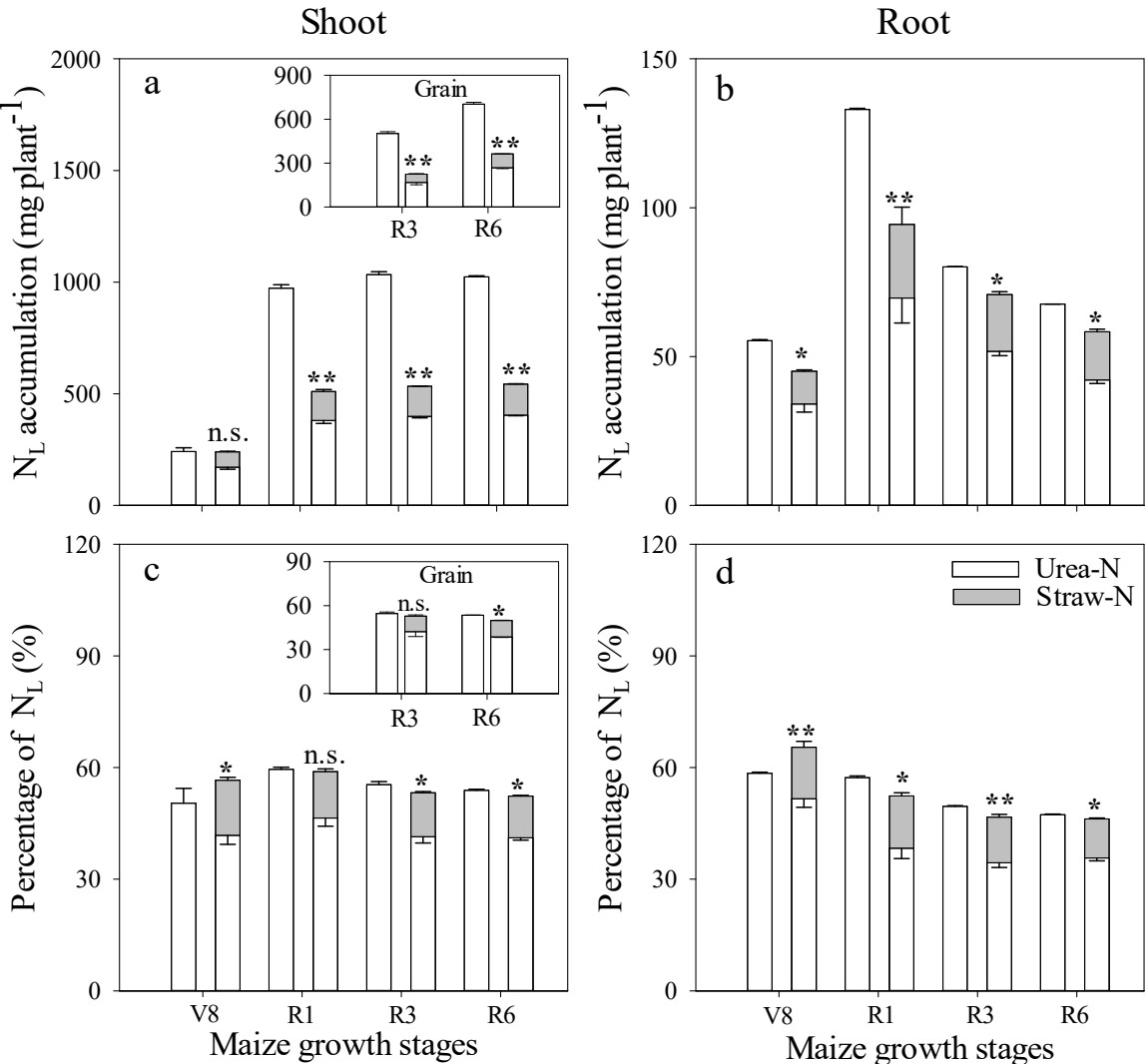

**Figure 2.** Concentration (**a**,**b**) and percentage (**c**,**d**) of labeled applied N (N_L) present as N taken up by maize shoots, grains, and roots in different fertilizer treatment during maize growth. White-only bars represent urea treatment (U). White plus gray bars represent urea combined with straw treatment (US). Maize growth stage: V8, R1, R3, and R6 respectively represent 8th leaf stage, silking stage, milk stage, and physiological maturity. Data shown are mean ± standard deviation of three replicates; * and ** denote significant differences at $p < 0.05$ and $p < 0.01$, respectively; n.s., not significant ($p > 0.05$). Shoot N includes leaf, stem, cob, husk, and grain fractions (all whole-plant fractions except roots).

### 3.2. Effect of Fertilizer Management on Applied N Distribution in Soil

Compared with the U treatment, the US treatment significantly ($p < 0.05$, Figure 3a–d) increased the concentration and percentage of applied N as soil TN by 24.0–253.0% and 28.1–242.1% in the bulk soil from stages V8 to R6 and increased those by 7.3–59.6% and 11.7–68.5% in rhizospheric soil from stages R1 to R6.

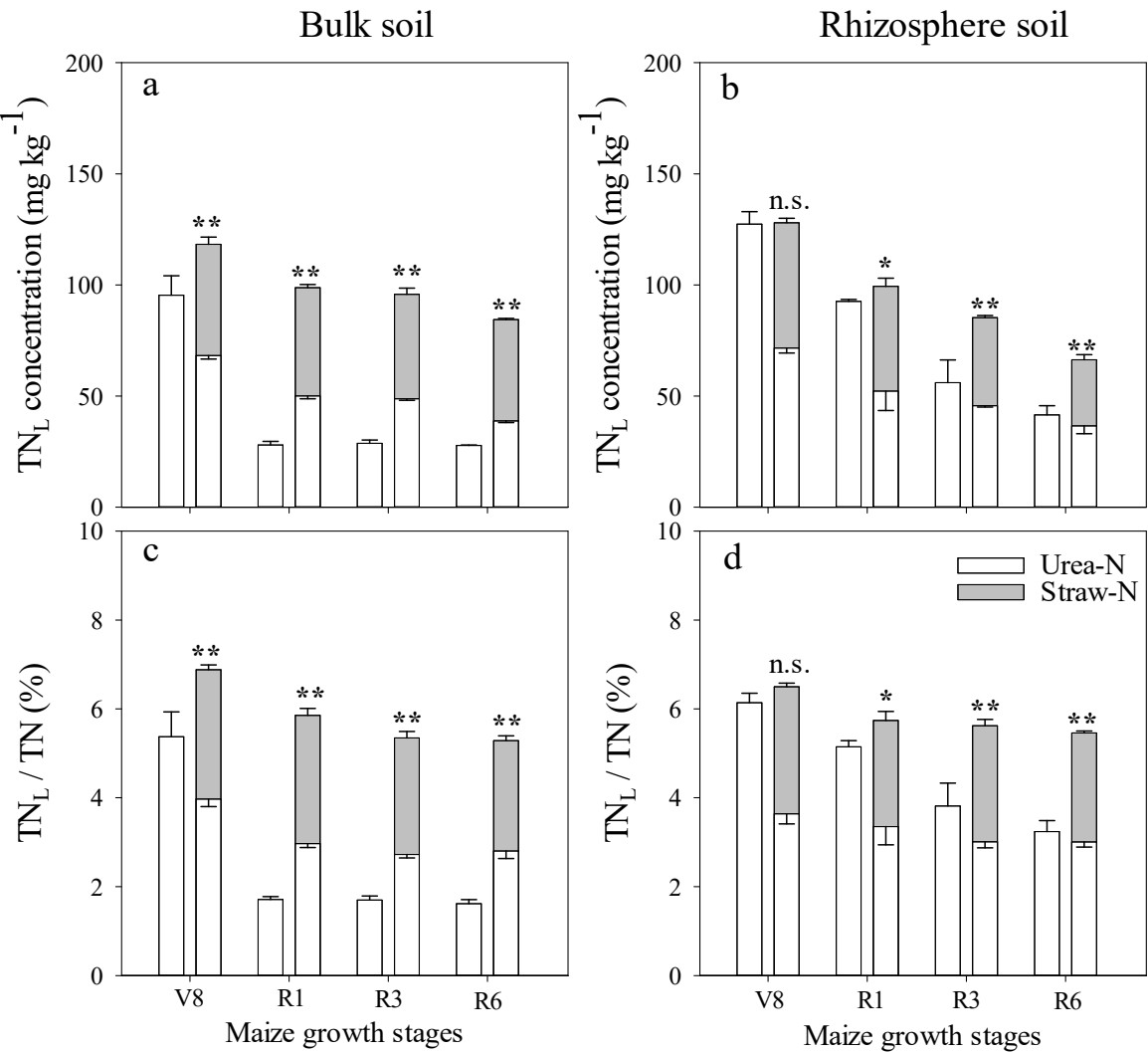

**Figure 3.** Concentration (**a**,**b**) and percentage (**c**,**d**) of labeled applied N ($N_L$) present as total N (TN) in bulk and rhizosphere soil in different fertilizer treatment during maize growth. White-only bars represent urea treatment (U). White plus gray bars represent urea combined with straw treatment (US). Maize growth stage: V8, R1, R3, and R6 respectively represent 8th leaf stage, silking stage, milk stage, and physiological maturity. Data shown are mean ± standard deviation of three replicates; * and ** denote significant differences at $p < 0.05$ and $p < 0.01$, respectively; n.s., not significant ($p > 0.05$).

The concentration and percentage of applied N as Inorg-N in the US treatment were significantly ($p < 0.05$, Figure 4a–d) 23.1–92.7% and 38.2–73.5% lower in bulk soil and 41.4–92.8% and 36.4–74.9% lower in rhizospheric soil than the U treatment from stages V8 to R1. The US treatment significantly ($p < 0.05$, Figure 4a–d) increased the concentration and percentage of applied N as Inorg-N at stage R3 in comparison with the U treatment.

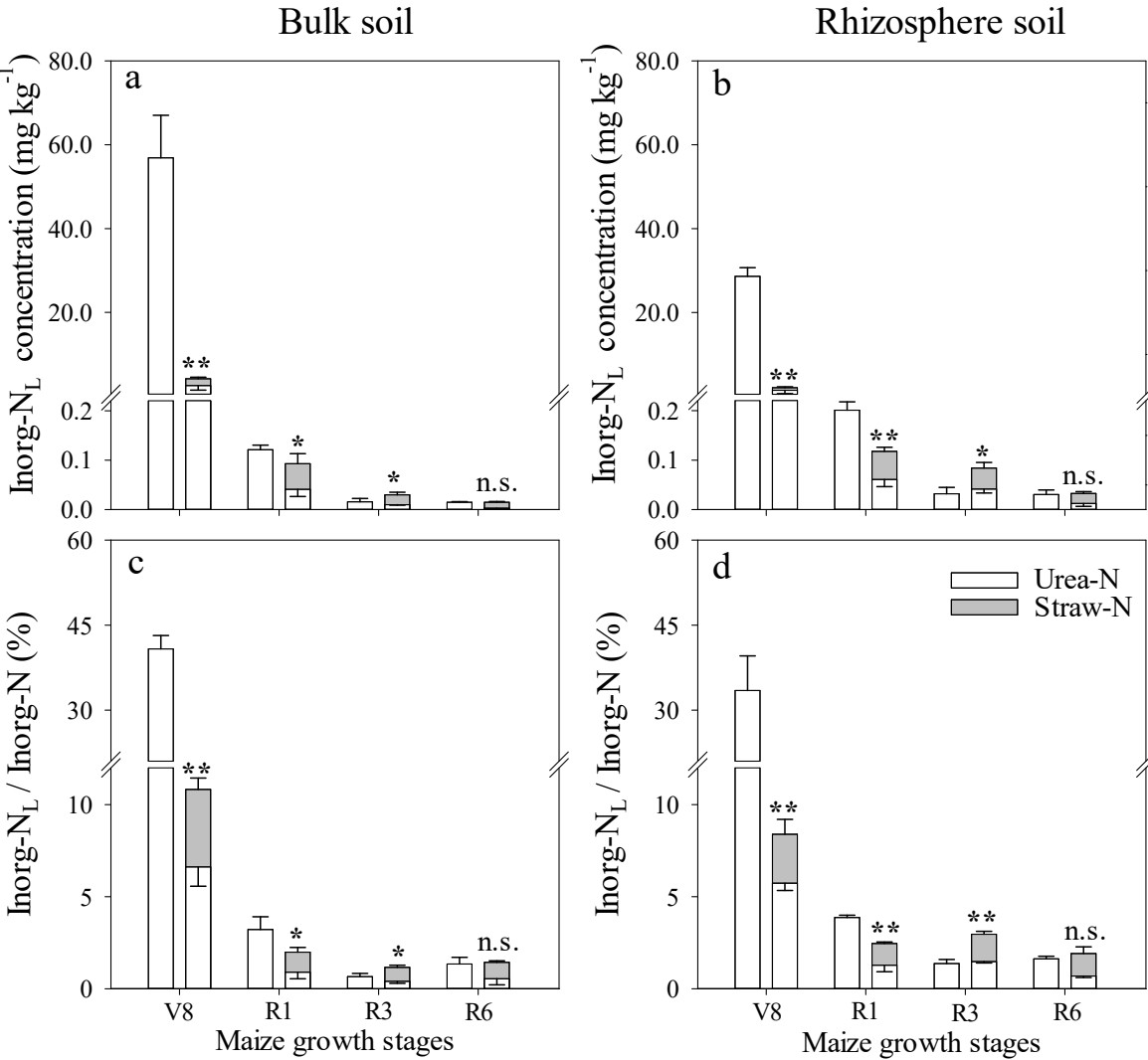

**Figure 4.** Concentration (**a**,**b**) and percentage (**c**,**d**) of labeled applied N ($N_L$) present as inorganic N (Inorg-N) in bulk and rhizosphere soil in different fertilizer treatment during maize growth. White-only bars represent urea treatment (U). White plus gray bars represent urea combined with straw treatment (US). Maize growth stage: V8, R1, R3, and R6 respectively represent 8th leaf stage, silking stage, milk stage, and physiological maturity. Data shown are mean ± standard deviation of three replicates; * and ** denote significant differences at $p < 0.05$ and $p < 0.01$, respectively; n.s., not significant ($p > 0.05$).

The US treatment significantly ($p < 0.05$, Figure 5a–d) increased the concentration and percentage of applied N as DON by 71.0–76.9% and 53.8–58.6% in bulk soil from stages R3 to R6 and increased those by 32.3–201.1% and 23.5–60.5% in rhizospheric soil from stages V8 to R6.

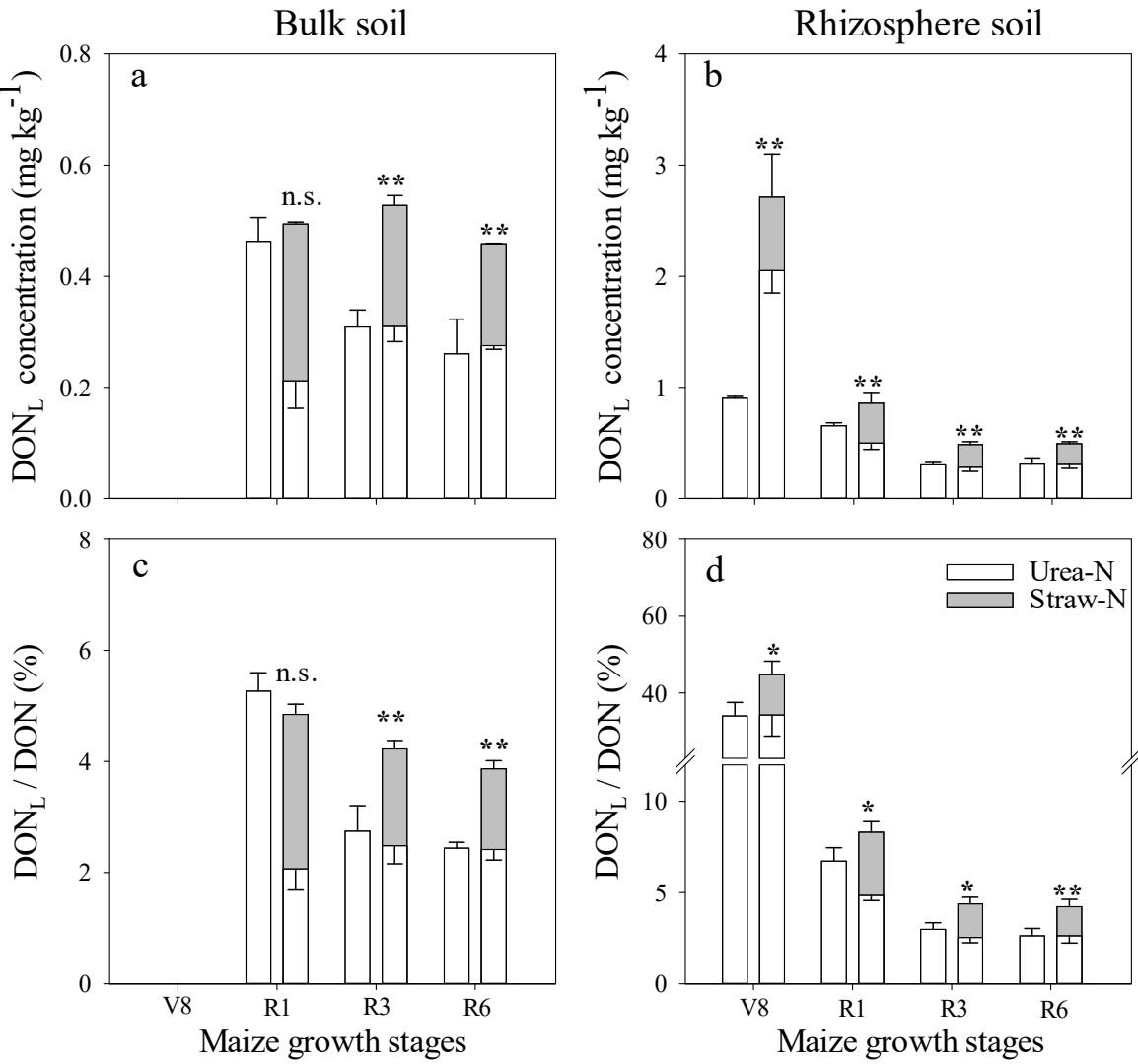

**Figure 5.** Concentration (**a**,**b**) and percentage (**c**,**d**) of labeled applied N (N$_L$) present as dissolved organic N (DON) in bulk and rhizosphere soil in different fertilizer treatment during maize growth. White-only bars represent urea treatment (U). White plus gray bars represent urea combined with straw treatment (US). Maize growth stage: V8, R1, R3, and R6 respectively represent 8th leaf stage, silking stage, milk stage, and physiological maturity. Data shown are mean ± standard deviation of three replicates; * and ** denote significant differences at $p < 0.05$ and $p < 0.01$, respectively; n.s., not significant ($p > 0.05$). Not detected in bulk soil at the V8 stage.

The concentration and percentage of applied N as MBN in the US treatment were significantly ($p < 0.05$, Figure 6a–d) 152.2–226.2% and 90.8–159.7% higher than the U treatment in bulk soil from stages R1 to R6 and 14.2–156.7% and 12.9–61.6% higher than the U treatment in rhizospheric soil from stages V8 to R3.

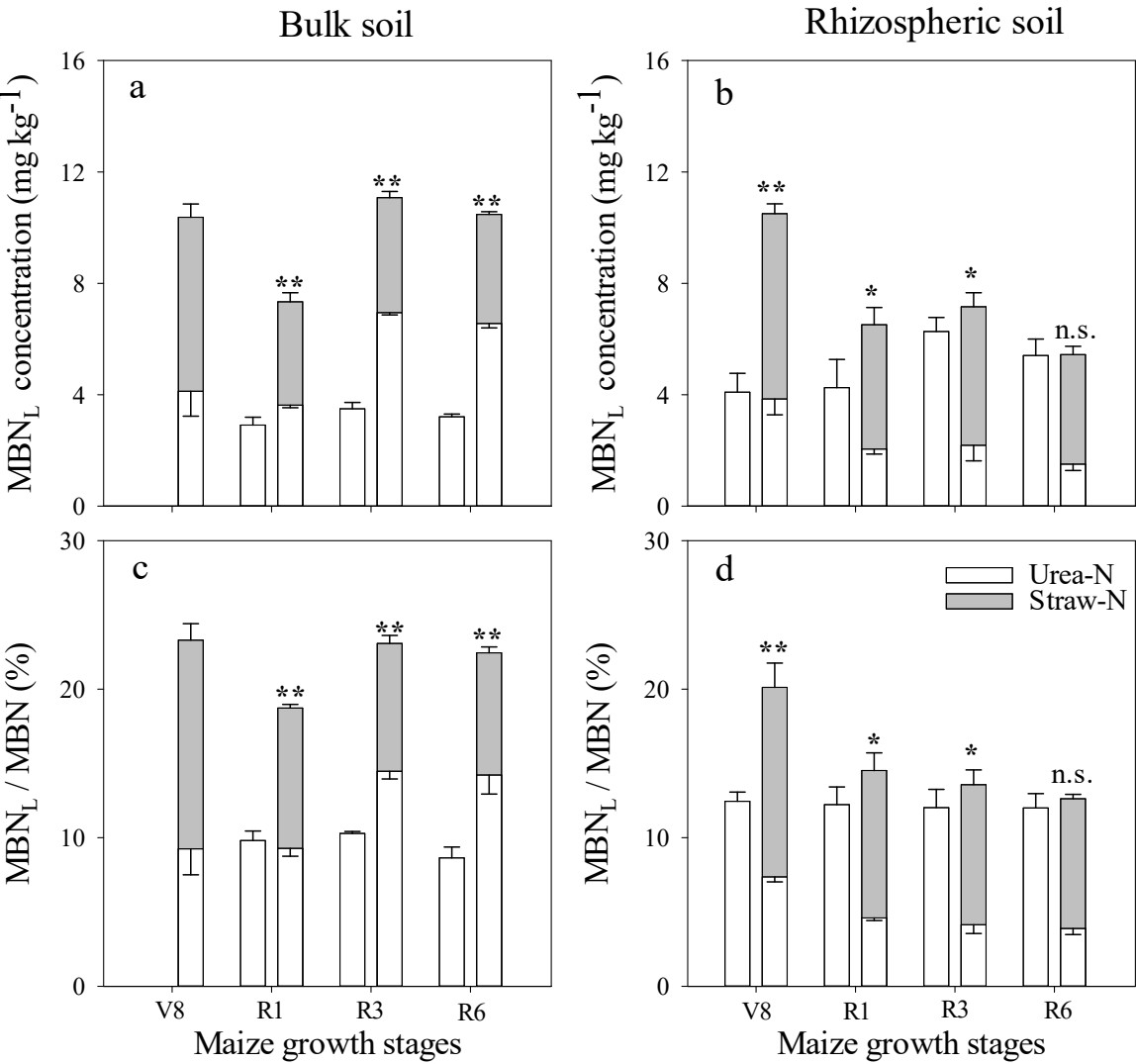

**Figure 6.** Concentration (**a,b**) and percentage (**c,d**) of labeled applied N (N$_L$) present as microbial biomass N (MBN) in bulk and rhizosphere soil in different fertilizer treatment during maize growth. White-only bars represent urea treatment (U). White plus gray bars represent urea combined with straw treatment (US). Maize growth stage: V8, R1, R3, and R6 respectively represent 8th leaf stage, silking stage, milk stage, and physiological maturity. Data shown are mean ± standard deviation of three replicates; * and ** denote significant differences at $p < 0.05$ and $p < 0.01$, respectively; n.s., not significant ($p > 0.05$).

Compared with the U treatment the US treatment significantly ($p < 0.01$, Figure 7a–d) increased the concentration and percentage of applied N as PON by 2.7–7.1 and 2.6–7.2 times in bulk soil and increased those by 1.8–3.6 and 1.7–3.1 times in rhizospheric soil. The US treatment also significantly ($p < 0.01$, Figure 8a–d) increased the concentration and percentage of applied N as MTN by 1.9–2.2 and 1.9–2.3 times in bulk soil and increased those by 0.3–1.6 and 0.4–1.3 times in rhizospheric soil.

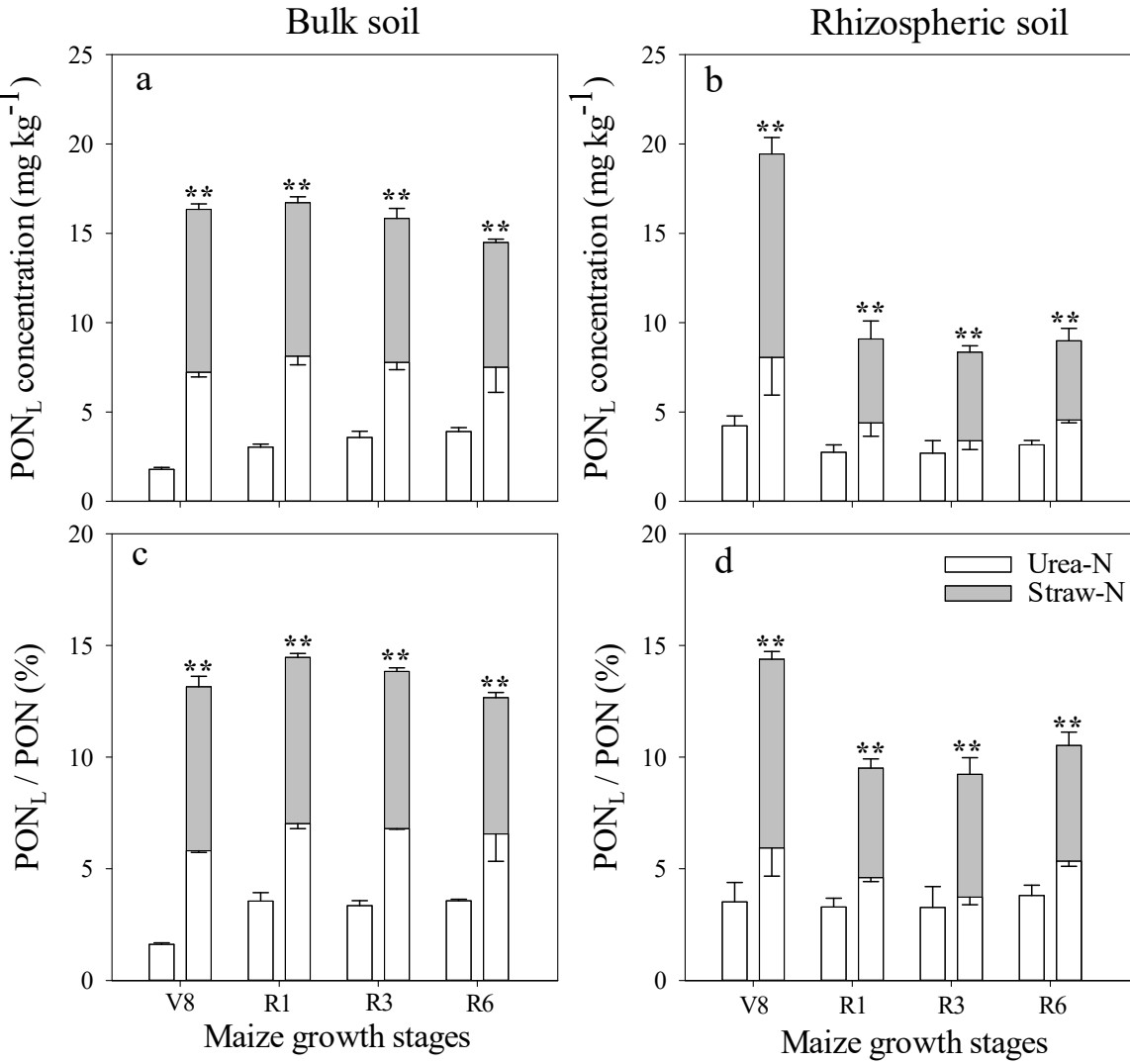

**Figure 7.** Concentration (**a**,**b**) and percentage (**c**,**d**) of labeled applied N ($N_L$) present as particulate organic N (PON) in bulk and rhizosphere soil in different fertilizer treatment during maize growth. White-only bars represent urea treatment (U). White plus gray bars represent urea combined with straw treatment (US). Maize growth stage: V8, R1, R3, and R6 respectively represent 8th leaf stage, silking stage, milk stage, and physiological maturity. Data shown are mean ± standard deviation of three replicates; ** denote significant differences at $p < 0.01$.

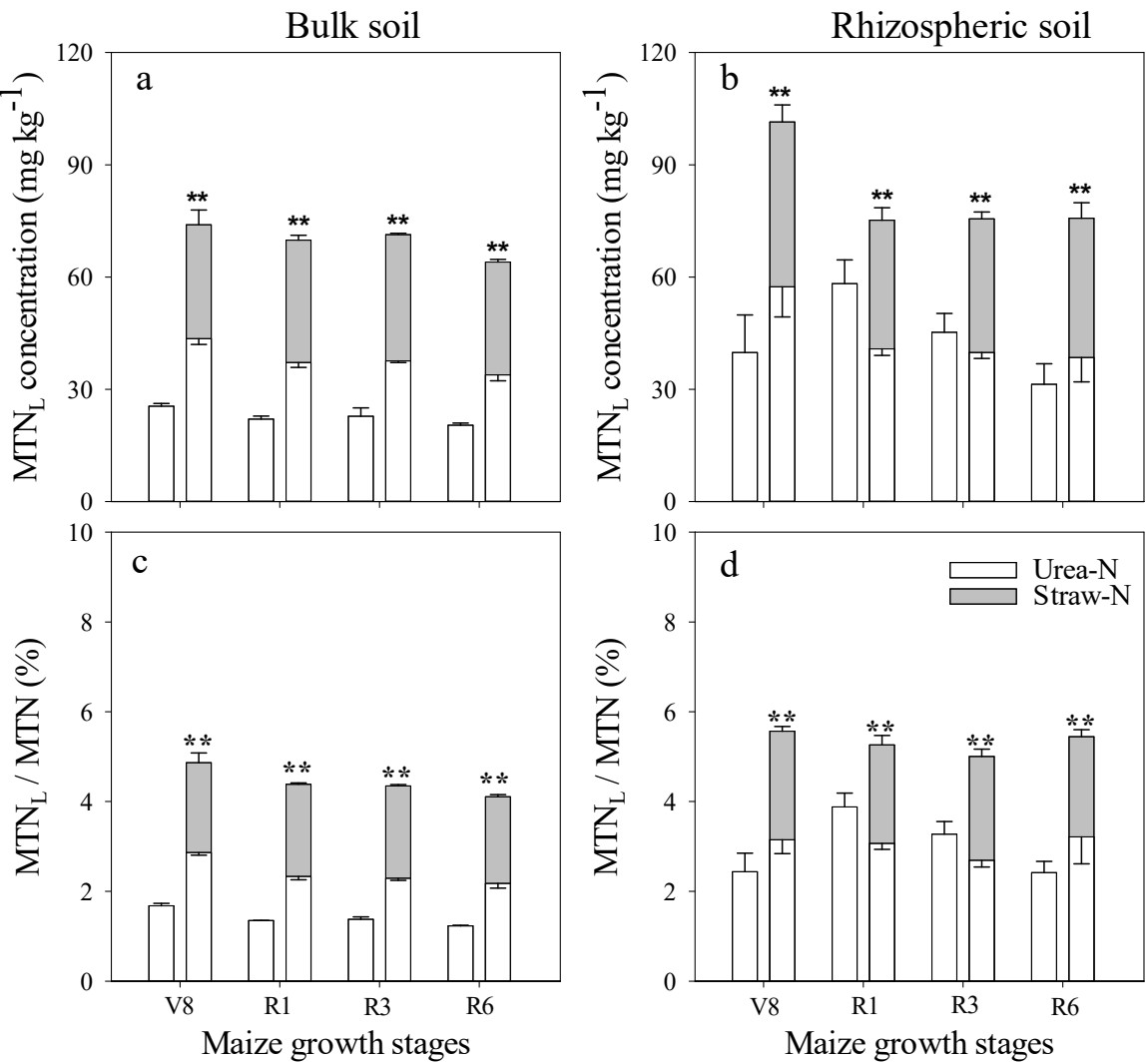

**Figure 8.** Concentration (**a**,**b**) and percentage (**c**,**d**) of labeled applied N ($N_L$) present as mineral associated total N (MTN) in bulk and rhizosphere soil in different fertilizer treatment during maize growth. White-only bars represent urea treatment (U). White plus gray bars represent urea combined with straw treatment (US). Maize growth stage: V8, R1, R3, and R6 respectively represent 8th leaf stage, silking stage, milk stage, and physiological maturity. Data shown are mean ± standard deviation of three replicates; ** denote significant differences at $p < 0.01$.

### 3.3. Comparison of N Distribution between Bulk and Rhizospheric Soils

Compared with bulk soil in the urea treatment, rhizospheric soil showed significant ($p < 0.05$, Table 2) increases in the concentration and percentage of urea N as TN, Inorg-N and MBN from stages R1 to R6 and MTN from stages V8 to R6. Conversely, rhizospheric soil showed decreased concentrations of urea N to PON from stages R1 to R6.

Compared with bulk soil in the US treatment, rhizospheric soil significantly ($p < 0.05$, Table 2) decreased the concentrations of applied N (urea N plus straw N) and urea N as TN from stages R3 to R6, decreased concentrations and percentages of applied N and urea N as MBN and PON from stages R1 to R6, and increased concentrations and percentages of applied N and urea N as Inorg-N from stages R1 to R6 and MTN from stages V8 to R6. The rhizospheric soil significantly ($p < 0.05$, Table 2) decreased the concentration of straw N as TN from stage R1 to R6, decreased the concentrations and percentages of straw N as PON from stages R1 to R6, and increased the concentrations and percentages of straw N as Inorg-N from stages R1 to R6 and MTN from stages V8 to R6.

**Table 2.** Statistical analysis of the concentration and percentage of applied N in different soil N pools between bulk soil and rhizospheric soil under urea (U) and urea plus straw (U + S) treatments at maize growth stages of 8th leaf (V8), silking (R1), milk (R3), and physiological maturity (R6) (data shown in Figures 2–8).

| Treatment | N Source | Stage | Concentration | | | | | | Percentage | | | | | |
|---|---|---|---|---|---|---|---|---|---|---|---|---|---|---|
| | | | TN | Inorg-N | DON | MBN | PON | MTN | TN | Inorg-N | DON | MBN | PON | MTN |
| U | Urea N | V8 | **, + | **, − | nd | nd | **, + | **, + | *, + | **, − | nd | nd | *, + | *, + |
| | | R1 | **, + | **, + | **, + | **, + | *, − | **, + | **, + | *, + | *, + | **, + | *, − | **, + |
| | | R3 | *, + | **, + | ns, − | **, + | **, − | **, + | **, + | **, + | ns, + | **, + | *, − | **, + |
| | | R6 | **, + | **, + | ns, + | **, + | *, − | *, + | **, + | **, + | ns, + | **, + | *, − | **, + |
| US | Urea N + Straw N | V8 | ns, + | **, − | nd | *, − | *, + | **, + | ns, − | *, − | nd | *, − | **, + | **, + |
| | | R1 | ns, + | **, + | **, + | *, − | **, − | **, + | ns, − | *, + | **, + | **, − | **, − | **, + |
| | | R3 | **, − | **, + | ns, − | **, − | **, − | *, + | ns, + | **, + | ns, + | **, − | **, − | *, + |
| | | R6 | **, − | **, + | ns, + | **, − | **, − | *, + | ns, + | *, + | ns, + | **, − | **, − | **, + |
| | Urea N | V8 | ns, + | **, − | nd | *, − | *, + | *, + | ns, − | *, − | nd | *, − | *, + | *, + |
| | | R1 | ns, + | *, + | **, + | **, − | **, − | **, + | ns, + | *, + | **, + | **, − | **, − | **, + |
| | | R3 | **, − | **, + | ns, − | **, − | **, − | *, + | ns, + | **, + | ns, + | **, − | **, − | *, + |
| | | R6 | **, − | **+ | ns, + | **, − | **, − | *, + | ns, + | *, + | ns, + | **, − | *, − | **+ |
| | Straw N | V8 | ns, + | *, − | nd | ns, + | *, + | **, + | ns, − | *, − | n.d. | ns, − | **, + | *, + |
| | | R1 | *, − | *, + | ns, + | ns, + | *, − | *, + | ns, − | *, + | ns, + | ns, + | **, − | *, + |
| | | R3 | *, − | *, + | ns, − | ns, + | **, − | *, + | ns, + | *, + | ns, + | ns, + | *, − | **, + |
| | | R6 | **, − | *, + | ns, − | ns, + | *, − | *, + | ns, + | **, + | ns, + | ns, + | *, − | **, + |

TN, soil total N; Inorg-N, inorganic N; DON, dissolved organic N; MBN, soil microbial biomass N; PON, soil particulate organic N; MTN, mineral associate total N. * and ** denote significant at $p < 0.05$ and $p < 0.01$, respectively; ns, not significant at $p < 0.05$. nd, no data. +, values higher in rhizosphere soil than in bulk soil; −, values lower in rhizosphere soil than in bulk soil.

### 3.4. Applied N Recovery in Plant and Soil N Pools

The recovery of urea N in the whole plant (mainly in the grain) was maximum in the U treatment, with only 18.5% urea N retained in the soil remaining mainly in the form of MTN, and accumulated urea N loss reached 10.0%. Straw addition significantly ($p < 0.5$, Table 3) reduced the recovery of urea N in the whole plant and N loss and increased urea N retained in the soil especially in the forms of PON and MTN. The recovery of straw N in the whole plant (26.0%) was significantly lower than that of urea N and most straw N was retained in the soil in the form of MTN (50.2%), with no loss of straw N.

**Table 3.** Recovery rate of applied N in plant and soil N pools under urea (U) and urea plus straw (U + S) treatments at maturity. Unit: %.

| Treatment | N Source | Plant Parts | | | Whole Plant | Soil Pools | | | | | Soil Total N | N Loss |
|---|---|---|---|---|---|---|---|---|---|---|---|---|
| | | Roots | Shoots | Grains | | Inorg-N (×10⁻²) | DON (×10⁻²) | MBN | PON | MTN | | |
| U | Urea | 4.5 ± 0.1 a | 67.0 ± 1.1 a | 46.9 ± 0.8 a | 71.5 ± 2.3 a | 1.2 ± 0.07 ab | 20.0 ± 0.4 b | 2.1 ± 0.05 c | 2.6 ± 0.2 c | 13.6 ± 0.4 c | 18.5 ± 0.8 c | 10.0 ± 1.1 a |
| US | Urea | 5.3 ± 0.2 a | 49.6 ± 2.1 b | 32.7 ± 1.7 b | 54.9 ± 1.7 b | 0.8 ± 0.06 b | 33.5 ± 2.8 a | 7.5 ± 0.8 a | 8.3 ± 1.6 b | 37.7 ± 1.8 b | 40.5 ± 1.1 b | 4.6 ± 0.5 b |
| | Straw | 2.7 ± 0.1 b | 23.3 ± 1.6 c | 15.7 ± 1.3 c | 26.0 ± 1.1 c | 1.9 ± 0.04 a | 32.3 ± 3.0 a | 6.5 ± 0.2 b | 11.7 ± 0.3 a | 50.2 ± 1.2 a | 74.9 ± 2.3 a | 0 |

Values are mean ± standard deviation of three replicates. Shoot N includes leaf, stem, cob, husk, and grain fractions. TN, soil total N; Inorg-N, inorganic N; DON, dissolved organic N; MBN, soil microbial biomass N; PON, soil particulate organic N; MTN, mineral associate total N; [15]NU, labeled urea-only; [15]NU + S, labeled urea + straw; [15]NS + U, labeled straw + urea; $N_L$, labeled fertilizer N. Values followed by different lower-letters in the same column for different treatments are significantly different at $p < 0.05$.

## 4. Discussion

### 4.1. Crop N Uptake

Generally, crop biomass is positively related to crop N uptake within the optimum N rate range [43,44]. In this equivalent N experiment, urea N immobilization in soil induced by straw and plenty of recalcitrant N in straw [30,45] in the US treatment significantly ($p < 0.05$) decreased shoot or grain biomass (Figure 1a) as well as N accumulation

and applied N percentage (Figure 2) from the R1 stage compared with the U treatment. The US treatment had a significantly ($p < 0.05$, Table 3) lower applied N recovery at maturity compared with the U treatment, significantly higher applied N percentage in shoots and roots ($p < 0.05$, Figure 2c,d) at stage V8, and no significant difference in applied N content in the shoots (Figure 2a). The significantly lower N content in the roots ($p < 0.05$, Figure 2b) in the US treatment indicates that the U treatment enhanced crop uptake of soil "native" N because the straw active C stimulated microbial activity and promoted the immobilization of the applied N and the re-immobilization of the mineralized "native" soil N [8]. This is also supported by the lower Inorg-N concentration in the US treatment (Figure 4a,b).

Nitrogen shortage can stimulate an increase in root biomass to acquire available N sources [46,47], thus the US treatment significantly (Figure 1b) increased root biomass from stages R1 to R3 in comparison with the U treatment. Root senescence as crop growth proceeded led to a decrease in root biomass [48,49] as shown by the roots from stage R1 in Figure 1b. Moreover, N remobilization occurs preferentially from belowground to sustain the aboveground N under low-N conditions [50], and both explanations above resulted in the accumulation of applied N in the roots showing a declining trend from stage R1 in both N application treatments (Figure 2b).

Straw N appeared from the V8 stage of maize in the US treatment in stover, grain, and roots at a range of 24.4–28.7% (Figure 2), and at maturity the straw N recovery was 26% (Table 3), indicating that straw N can be assimilated by the crop and similar to results of Li et al. [51], this confirmed our hypothesis that a portion of straw N was assimilated by the crop. However, the lower shoot biomass and grain in the US treatment than the U treatment (Figure 1) indicates that some of the straw N was unavailable to the crop.

### 4.2. Soil N Pools

Straw incorporation increased the soil retention of applied N and promoted soil N transformation. On one hand, part of the recalcitrant N in straw was unavailable to microbes and the crop, resulting in a larger amount of straw N detained in the soil [52]. On the other hand, the labile C and N sources in straw provided energy for microbial N immobilization [53–55]. Hence, the US treatment significantly ($p < 0.05$) decreased the concentrations and percentages of applied N as Inorg-N from stages V8 to R1 (Figure 4) and urea N loss (Table 3), and significantly ($p < 0.05$) increased the concentrations and percentages of urea N as TN and MBN at most growth stages compared with the U treatment in both bulk and rhizospheric soils (Figures 3 and 6). At stage V8, the maize was in rapid vegetative growth stage, both inorganic N and low-molecular-weight compounds can be assimilated by crops [56,57], and the shortage of available N resulting in competition for N between maize and microorganisms may be responsible for the disappearance of applied N in the DON in both treatments and in MBN in the U treatment in the bulk soil (Figures 5a,b and 6a,b) [17].

Straw was an important component of soil particulate organic matter [58,59] and decomposed straw N and chemical fertilizer N can be adsorbed onto soil mineral particles (<53 μm) with a high specific surface area [60]. Thus, relative to the U treatment, the US treatment significantly ($p < 0.01$, Figure 7a,c) increased the concentrations and percentages of PON and MTN and the markedly higher concentration and percentage of straw N at each growth stage. Particulate organic matter is an unprotected soil fraction that provides C substrates for microorganisms and thus promotes microbial turnover [18,21]. The N from microbial residues or low-molecular-weight compounds from decomposable straw is further associated with <53 μm soil particles around particulate organic matter [21], therefore the US treatment increased the concentrations and percentages of urea N as PON by 25.6–305.6% and 14.1–260.9% in the bulk and rhizospheric soils relative to the U treatment (Figure 7). Immobilization of chemical fertilizer N by microorganism and adsorption by <53 μm fractions [27,] resulted in increases in the concentrations and percentages of urea N as MTN by 64.7–70.9% and 65.9–76.4% in the US

treatment in comparison with the U treatment in bulk soils (Figure 8a,c). The contribution of straw N to the concentration of MTN was 41.1–49.1% from growth stages V8 to R6.

Straw N entered different soil N pools and straw decomposition was microbially driven [56], as shown in Figures 3–8. In different N pools during maize growth (Figures 3–8) the decrease in straw N in PON in rizhospheric soil was maximum (Figure 7) because the straw was "temporal" particulate organic matter in the soil and the N from decomposable particulate organic matter was transformed to the other N pools by microorganisms [32,59].

*4.3. Comparison between Bulk and Rhizospheric Soils*

The rhizosphere is a zone of intense microbial activity [52] and the continuing N uptake by maize promotes N transport from bulk to rhizospheric soil [57]. This may be explained by the U treatment having the higher concentrations and percentages of applied N as TN from stages V8 to R6 and as MBN from stages R1 to R6 in rhizospheric soil than in bulk soil (Table 2). With crop N uptake under the relatively low available N rate in the US treatment, the immobilization of chemical fertilizer N and the recalcitrant N in straw due to straw application limited N transport from bulk to rhizospheric soil [54,55], thus the rhizospheric soil in the US treatment significantly ($p < 0.05$, Table 2) decreased the concentrations of applied N as TN from stages R3 to R6, and the concentrations and percentages of applied N as MBN from stages R1 to R6 in comparison with bulk soil. Compared to the bulk soil in the US treatment, the significantly lower concentrations and nonsignificant percentage of applied N or urea N to soil TN in rhizospheric soil from stages R3 to R6 suggested that rhizodeposition triggered microbial activity and accelerated the mineralization of soil native N to meet crop N demand under the low N availability situation [58]. Moreover, the decomposition of straw N also increased in rhizospheric soil in the US treatment as shown by the higher percentages of straw N in Inorg-N and MBN and the lower values in PON and DON in rhizospheric soil from stage R1 overall (Figures 4–8), this supported our hypothesis that the rhizosphere accelerated the transformation of organic N. The Mollisol is dominated by 2:1 type soil clay minerals [61] and low-molecular-weight compounds from rhizodeposition can restrain $NH_4^+$ diffusion from interlayer of soil minerals to the soil solution [9]. Furthermore, part of the microbial residue associated with soil mineral particles was not reutilized by microorganisms [30], and this may have resulted in the significantly ($p < 0.05$, Table 2) higher concentrations and percentages of applied N as MTN in both N application treatments in the rhizospheric soil compared with the bulk soil.

**5. Conclusions**

The results indicated that a substantial amount of straw recalcitrant N in the US treatment suppressed maize yields at the equivalent N rate despite the 26.0% straw N recovery in maize at maturity. The combination of chemical fertilizer and straw decreased the soil inorganic N content and increased the immobilization of chemical fertilizer N or the transformation of straw N to different soil N pools, and limited N flow from the bulk soil to the rhizospheric soil. However, the rhizosphere promoted the bioavailability of straw N and the transformation of chemical fertilizer N and straw N from PON to MTN. In general, the results contributed to our understanding of the transformation of different N sources in the soil–crop system under the combined application of chemical fertilizer and straw and the N regulatory effect of the rhizosphere on different N sources requires further consideration in farming practice.

**Author Contributions:** Data analysis, Writing, and Revision, J.Z.; Experimental condition support, Review, P.H., D.W., L.J., L.Z., L.L., S.Z., X.X., W.Z.; Supervision, L.Z.; Experimental design, Data analysis, Project administration, Funding acquisition, Revision, S.Q. All authors have read and agreed to the published version of the manuscript.

**Funding:** This research was funded by the National Natural Science Foundation of China, grant number 41101277; the National Key Research and Development Plan, grant number 2018YFD0200804, 2016YFD0200101, and 2018YFD0201001; the Fundamental Research Funds for Central Non-Profit Scientific Institutions, grant number 1610132019014.

**Institutional Review Board Statement:** Not applicable.

**Informed Consent Statement:** Not applicable.

**Data Availability Statement:** Relevant data applicable to this research are within the paper.

**Acknowledgments:** We are very grateful for the valuable suggestions and modifications made by Peter Christie in this research work.

**Conflicts of Interest:** The authors declare no conflict of interest.

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
