# Peer review of "Changes in Nitrogen Pools in the Maize–Soil System after Urea or Straw Application to a Typical Intensive Agricultural Soil: A 15N Tracer Study"

_agronomy, doi:10.3390/agronomy11061134_

Round 1
Reviewer 1 Report
The authors did an interesting experiment to compare the N value of fertilizer alone and fertilizer combined with straw at different scales (N soil pools, N recovery, rhizosphere vs bulk). I have indicated a few points throughout the intro, M&M and results that require attention and various points in the discussion that need a better explanation before accepting the manuscript.
L47: what do you mean by energy? Define
L47: substrates -> sources
L48: and this will affect -> affecting
L 48-50: reformulate the sentece
L54: delete when -> large amounts of readily available C derived from root exudates in the rhizosphere promote microbial N immobilization, which contributes to the increasing competition for N
L 60: this transport is closely correlated with N formation: what do you mean?
L67: define better the labile pool
L 73: delete and
L 77:what do you mean when PON is removed?
L92: rephrase this last sentence. There is much research about the contribution of chemical fertilizers to soil N pool.
L 94: rephrase. …quantified the effect
L 99: rephrase. It is not correct to say The rhizosphere will accelerate… the transformation would be accelerated in the rhizosphere
L 108: typo
Table 1: formatting
L117:methodology for N limitation overcome is not clear to me
L128: how is it possible to have a 0.37% of 15N in the unlabeled straw? How do you explain the differences in total N between the label and unlabeled straw?
L133: how did you label the urea?
L 146: it is not clear why would you put 3 PVC tubes in the pot. Do the roots then grow in the wall of the PVC? Do you have a picture that illustrates the setup?
L 213: the aim of showing two different calculations is not clear to me. Why not focus only on the concentrations of applied N that are recovered in the plant or appear in the soil N pools?
L250: move the percentages at the end of the sentence. Also, fig 2a and b are not showing % but mg plant-1
L251: % of N uptake from the applied N you mean?
L256: if you show only one of the calculations, you could have Fig 1 and 2 combines.
L272: there are too many figures. A combination of Fog 3, 4, 5 could be done. Reduce the calculations to only 1.
L369: I miss a discussion of why you chose to measure different growth stages of the maize plant. There is no conclusion related to differences between treatments on the time of growth. I miss a discussion of why the initial hypothesis where a combined application of chemical fertilizer and straw at controlled rates would be an efficient method for increasing soil nutrient status was, in the end, not accepted. I consider interesting the hypothesis that the straw active C stimulated microbial activity and promoted the immobilization of the applied N and the re-immobilization of the mineralized “native” soil N. However, there is no direct measure of the microbial activity that you can link it to your suggestion. You mentioned it is supported by the lower Inorg-N concentration in the US treatment but not by the MTN?
Also, in the discussion, I miss the point justifying the use of 15N and explaining why it might be more attractive based on the results than standard N balances studies.
L464: you start here a discussion point related to manure application, but you did not include this in your study, and therefore no link to the Figs and Tables should be added.
Author Response
Response to Reviewer 1 Comments
The authors did an interesting experiment to compare the N value of fertilizer alone and fertilizer combined with straw at different scales (N soil pools, N recovery, rhizosphere vs bulk). I have indicated a few points throughout the intro, M&M and results that require attention and various points in the discussion that need a better explanation before accepting the manuscript.
L47: what do you mean by energy? Define
Response: This word has been deleted.
L47: substrates -> sources
Response: This word has been modified according reviewer’s suggestion, please see L.55.
L48: and this will affect -> affecting
Response: This sentence has been modified according reviewer’s suggestion, please see L.55.
L 48-50: reformulate the sentence
Response: This sentence has been revised, please see L.56-58.
L54: delete when -> large amounts of readily available C derived from root exudates in the rhizosphere promote microbial N immobilization, which contributes to the increasing competition for N
Response: This sentence has been deleted according to reviewer’s suggestion.
L 60: this transport is closely correlated with N formation: what do you mean?
Response: This sentence has been revised, please see L.66-67.
L67: define better the labile pool
Response: This sentence has been revised, please see L.74-75.
L 73: delete and
Response: this word has been deleted.
L 77:what do you mean when PON is removed?
Response: This sentence has been revised, please see L.84-85.
L92: rephrase this last sentence. There is much research about the contribution of chemical fertilizers to soil N pool.
Response: This sentence was deleted.
L 94: rephrase. …quantified the effect
Response: This sentence has been modified according to reviewer’s suggestion. Please see L.101.
L 99: rephrase. It is not correct to say The rhizosphere will accelerate… the transformation would be accelerated in the rhizosphere
Response: This sentence has been modified according to reviewer’s suggestion. Please see L.106-107.
L 108: typo
Response: This sentence has been revised, please see L.115-117.
Table 1: formatting
Response: The formatting of Table 1 has been revised. Please see Table 1.
L117:methodology for N limitation overcome is not clear to me
Response: This sentence has been revised, please see L.125.
L128: how is it possible to have a 0.37% of 15N in the unlabeled straw? How do you explain the differences in total N between the label and unlabeled straw?
Response: 15N natural abundance was 0.3663%, and 15N contained in soil and fertilizer could be assimilated by maize, which resulted in a 0.37% of 15N in the unlabeled straw. The differences in total N between the label and unlabeled straw was probably attributed that soil fertility was not homogeneous.
L133: how did you label the urea?
Response:15N labeled urea were produced by Shanghai Research Institute of Chemical Industry in China. And this information has been added in the revised manuscript, please see L.149-150.
L 146: it is not clear why would you put 3 PVC tubes in the pot. Do the roots then grow in the wall of the PVC? Do you have a picture that illustrates the setup?
Response: The pot experiment has a limited volume. In order to prevent roots root growth along the inner wall and compare N changes in bulk soil and rhizosphere soil, three PVC tubes were inserted vertically into the soil (as the following picture). The maize roots were mainly concentrated among the PVC tubes according to maize sampling at different stages.
L 213: the aim of showing two different calculations is not clear to me. Why not focus only on the concentrations of applied N that are recovered in the plant or appear in the soil N pools?
Response: Due to both urea and straw were labeled in this research, it offered us an opportunity to explore the effects of chemical fertilizer application with or without straw on native soil N and applied N. In order to understand whether there was a similar impact on applied N and native soil N, the percentage of applied N in different N pools was also calculated in this paper. Besides, the further explanation of the percentage of applied N was added in the manuscript, please see L.231-234.
L250: move the percentages at the end of the sentence. Also, fig 2a and b are not showing % but mg plant-1
Response: Sorry, the meaning of this sentence was not expressed clearly, and it has been revised. The meaning of this sentence was to state that ” The accumulation of applied N in plant shoots, grains, and roots in the US treatment significantly increased by 46.9%-48.4%, 48.4%-55.2%, and 24.6%-27.7% relative to the U treatment”. Please see L.257-259.
L251: % of N uptake from the applied N you mean?
Response: It’s not “percentage of N uptake from the applied N” but “the percentage of applied N” (i.e. the ratio of the concentration of applied N contained in plant to the concentration of total N contained in plant), and the calculated formation was presented in L.227, and the further explanation of the percentage of applied N was added in the manuscript, please see L.231-234.
L256: if you show only one of the calculations, you could have Fig 1 and 2 combines.
Response: Both of these two calculations are very important for this paper. The concentration of applied N could directly reflect the effect of fertilizer treatments on the distribution of applied N among plant and soil N pools, and the percentage of applied N could reflect whether there was a similar impact on applied N and native soil N. Thus, we have to adopt both of these two calculations.
L272: there are too many figures. A combination of Fog 3, 4, 5 could be done. Reduce the calculations to only 1.
Response: This experiment design offered us an opportunity to explore the effect of chemical fertilizer application with or without straw on applied N and native soil N distribution. Like the response mentioned above, both of these two calculations are very important for this paper. Thus, we have to adopt both of these two calculations.
L369: I miss a discussion of why you chose to measure different growth stages of the maize plant. There is no conclusion related to differences between treatments on the time of growth.
Response: Maize roots activities varied with growth stage, and roots activities greatly affected N uptake of crop and root secretions, which was bound to affect soil N transformation, so we chose to measure different growth stages of the maize plant. The reason for measure different growth stages of the maize plant was added in the revised manuscript, please see L.153-154.
Besides, in the discussion, the concentration of applied N in maize and soil N pools varied with growth period was discussed, for example, “…. as well as N accumulation and applied N percentage (Fig. 2) from the R1 stage compared with the U treatment” in L.331-332, “….. percentages of applied N as Inorg-N from stages V8 to R1 (Fig. 4) and urea N loss” in L.361-362, “…… of applied N as TN from stages V8 to R6 and of MBN from stages R1 to R6 in rhizospheric soil than in bulk soil” in L.395-396.
I miss a discussion of why the initial hypothesis where a combined application of chemical fertilizer and straw at controlled rates would be an efficient method for increasing soil nutrient status was, in the end, not accepted.
Response: Sorry, I don’t quite understand about your question. Firstly, there was not a hypothesis about that a combined application of chemical fertilizer and straw at controlled rates would be an efficient method for increasing soil nutrient status in the initial hypothesis;
Secondly, the recovery rate of applied N at maturity could reflect soil N status, and a combined application of chemical fertilizer and straw significantly increased applied N retention in soil in comparison with urea treatment, so we agree that a combined application of chemical fertilizer and straw at controlled rates would be an efficient method for increasing soil nutrient status;
Thirdly, this research mainly concerned the effects of chemical fertilizer application with or without straw on applied N and soil native N distribution in crop-soil system.
I consider interesting the hypothesis that the straw active C stimulated microbial activity and promoted the immobilization of the applied N and the re-immobilization of the mineralized “native” soil N. However, there is no direct measure of the microbial activity that you can link it to your suggestion. You mentioned it is supported by the lower Inorg-N concentration in the US treatment but not by the MTN?
Response: MTN was a stable N pool which was not easily accessible for soil microorganisms; besides, the size of MTN also could be regulated through abiotic pathway (e.g. sorption and desorption). The concentration of Inorg-N could indirectly reflect microbial activity in some degree due to that the release and immobilization of Inorg-N was mainly through biotic pathway.
Also, in the discussion, I miss the point justifying the use of 15N and explaining why it might be more attractive based on the results than standard N balances studies.
Response: Sorry, I don’t quite understand about your question. In the discussion, there was no point indicating that it might be more attractive based on the results than standard N balances studies. By contrast, standard N balances method was adopted in this research to calculated applied N loss.
In this pot experiment, there was no N leaching, and the limited pot space caused a higher N uptake of crop roots. Thus, the recovery rate of applied N in this experiment was more attractive.
L464: you start here a discussion point related to manure application, but you did not include this in your study, and therefore no link to the Figs and Tables should be added.
Response: There was no close relation between the discussion points related to manure application, so this paragraph was deleted.
Submission Date
22 April 2021
Date of this review
18 May 2021 13:35:26

Reviewer 2 Report
It is well-written paper on N transformations under different fertilization treatments for maize crop. The data are novel and interesting and merit publication. Some points below are proposed for amelioration/clarification.
Paragraph 2.3.:
It is not clear how the sub-treaments were performed; the US treatment had two sub-treatments or the U treatment? Please, explain about the sub-treatments.
It is not clear to me why you have chosen KCl instead of K2SO4 for fertilization. Didn't you afraid of Cl toxicity for the crop?
Paragraph 2.6.: Did you follow ONE or TWO-WAY ANOVA for your statistical analysis?
Recommednation to the Editor: Acceptable, after a satisfactory minor revision.
Author Response
It is well-written paper on N transformations under different fertilization treatments for maize crop. The data are novel and interesting and merit publication. Some points below are proposed for amelioration/clarification.
Paragraph 2.3.:
It is not clear how the sub-treaments were performed; the US treatment had two sub-treatments or the U treatment? Please, explain about the sub-treatments.
Response: More detail information about the sub-treatment was added in the part of experiment design. Please see L.141-145.
It is not clear to me why you have chosen KCl instead of K2SO4 for fertilization. Didn't you afraid of Cl toxicity for the crop?
Response: The reason for why we chosen KCl instead of K2SO4 for fertilization was that KCl was more popular than K2SO4 in local area, meanwhile, KCl was more soluble than K2SO4. Besides, according to farmer’s practice, the amount of Cl- application in current experiment was not enough to threaten the crops growth.
Paragraph 2.6.: Did you follow ONE or TWO-WAY ANOVA for your statistical analysis?
Response: One-way ANOVA statistical analysis was adopted in this research. And this information was also added in the part of 2.6, please see L.240-241.
Recommednation to the Editor: Acceptable, after a satisfactory minor revision.
Submission Date
22 April 2021
Date of this review
17 May 2021 14:14:32
